

# Transcriptomics analysis of field-droughted pear (*Pyrus spp.*) reveals potential drought stress genes and metabolic pathways

Sheng Yang[1,2], Mudan Bai[1,2], Guowei Hao[1], Huangping Guo[1] and Baochun Fu[1,2]

[1] Pomology Institute, Shanxi Agricultural University, Taiyuan, Shanxi, China
[2] Shanxi Key Laboratory of Germplasm Improvement and Utilization in Pomology, Taiyuan, Shanxi, China

## ABSTRACT

Drought acts as a major abiotic stress that hinders plant growth and crop productivity. It is critical, as such, to discern the molecular response of plants to drought in order to enhance agricultural yields under droughts as they occur with increasing frequency. Pear trees are among the most crucial deciduous fruit trees worldwide, and yet the molecular mechanisms of drought tolerance in field-grown pear remain unclear. In this study, we analyzed the differences in transcriptome profiles of pear leaves, branches, and young fruits in irrigation *vs* field-drought conditions over the growing seasons. In total, 819 differentially expressed genes (DEGs) controlling drought response were identified, among which 427 DEGs were upregulated and 392 DEGs were downregulated. Drought responsive genes were enriched significantly in monoterpenoid biosynthesis, flavonoid biosynthesis, and diterpenoid biosynthesis. Fourteen phenylpropanoid, five flavonoid, and four monoterpenoid structural genes were modulated by field drought stress, thereby indicating the transcriptional regulation of these metabolic pathways in fruit exposed to drought. A total of 4,438 transcription factors (TFs) belonging to 30 TF families were differentially expressed between drought and irrigation, and such findings signal valuable information on transcriptome changes in response to drought. Our study revealed that pear trees react to drought by modulating several secondary metabolic pathways, particularly by stimulating the production of phenylpropanoids as well as volatile organic compounds like monoterpenes. Our findings are of practical importance for agricultural breeding programs, while the resulting data is a resource for improving drought tolerance through genetic engineering of non-model, but economically important, perennial plants.

## INTRODUCTION

Along with an increasing global population, drought is becoming one of the most persistent factors that limits agricultural production and food security around the world, especially in arid and semi-arid regions (*Mittler, 2006*). In turn, drought is responsible for losses in the multibillions of dollars annually (*Fahad et al., 2017*; *Lesk, Rowhani & Ramankutty, 2016*). China is facing a perilous water crisis in which 50% of the national territory is located

Corresponding author
Baochun Fu, sxyyfbc@sxau.edu.cn

in arid and semi-arid regions (*Hu & Zhang, 2001*). The temporal-spatial distribution of annual precipitation causes 26.7% of the national land territorial area in Northwest China to have arid and semiarid climates, a region where drought is common. Predicting drought severity is difficult, and to do so requires consideration of several factors such as rainfall amount and distribution, evaporative demands, and the moisture storing ability of soils (*Saud et al., 2017*; *Tadesse & Melkam, 2016*). Globally, several management strategies have been implemented for improved crop production under drought environments (*Bodner, Nakhforoosh & Kaul, 2015*; *Fahad et al., 2017*). Among these, the development of crop varieties with an increased tolerance to drought functions as an important and effective strategy to combat drought.

Plants cope with water deficiency by complex mechanisms from molecular, biochemical and physiological processes at the cellular or whole plant level (*Bray, 1997*; *Goufo et al., 2017*; *Huber & Bauerle, 2016*; *Ruggiero et al., 2017*; *Zandalinas et al., 2017*). With the advent of new high throughput "-omics" technologies like proteomics and transcriptomics, notable strides have been made towards understanding the molecular mechanisms that regulate tolerance to drought. Previous studies have demonstrated signal transduction of drought stress perception to the nucleus *via* complex cellular signaling networks involving second messengers. These include reactive oxygen intermediates (ROIs) and calcium, calcium-associated proteins, and kinase cascade such as mitogen-activated protein (MAP) (*Bray, 1997*; *Chen et al., 2002*; *Huber & Bauerle, 2016*; *Knight & Knight, 2001*; *Liu et al., 1998*; *Zandalinas et al., 2017*). Drought stress signaling cascades are comprised of many stress-responsive genes. These include molecular chaperones such as late embryogenesis abundant (LEA) proteins and heat shock proteins (HSPs) that function as effector molecules. Other examples include transcription factors (TFs) like members of the APETALA2/ethylene-responsive element binding protein (AP2/EREPB), a basic leucine zipper (bZIP), WRKY, and MYB proteins that act as regulator molecules (*Shinozaki & Yamaguchi-Shinozaki, 2007*; *Song et al., 2005*; *Wang, Vinocur & Altman, 2003*). The physiological and molecular mechanisms of plant responses to drought have been extensively studied in model plants with dehydration treatments in controlled laboratory or greenhouse conditions (*Li, Xu & Huang, 2016*; *Wang et al., 2018*; *Zarafshar et al., 2014*). However, results from these studies most often translate poorly to field-grown plants. Clarifying the molecular mechanisms that regulate drought tolerance from crops grown under field conditions will facilitate a more thorough grasp of the complex interactions between drought response and environmental factors that crops encounter in the field during the growing season. As such, the task of developing an improved understanding of molecular elements in responsiveness to field drought in non-model plants will aid in both traditional and modern breeding applications towards improving stress tolerance.

Pear is one of the most vital fruit crops in the world and the second major crop among deciduous fruits in China after apples (*Silva et al., 2014*). The crop has considerable value both economically and in terms of personal health. In China, pear is primarily grown in the Northwestern region, accounting for 60 percent of pear production in the country. YuluXiangli (*Pyrus spp*) is an improved pear cultivar that is highly tolerant to drought, and it is an ideal source for examining genomic responses to drought in order to explore valuable

tolerance genes (*Okubo & Sakuratani, 2000*; *Zong et al., 2014*). The full genome sequencing and resequencing of multiple pear cultivars (*Huang et al., 2015*; *Li, Xu & Huang, 2016*; *Wang et al., 2018*; *Wu et al., 2013*) have enabled several transcriptome studies of drought responses in pear, thereby revealing a broad, multifaceted response to drought. Such a response features coordination between phytohormone signaling pathways, the reduction of photosynthetic gene expression, and the alteration in expression of genes involved in stress-induced leaf senescence. These studies, however, have been restricted to greenhouses under certain durations of drought stimuli treatment as opposed to field conditions that use early time points with samples exclusively from leaves (*Li, Xu & Huang, 2016*; *Wang et al., 2018*).

The primary objectives of the present study were to identify differentially expressed genes (DEGs) and to compare the gene expression patterns in leaf, branch, and fruit tissue of pear in response to drought induced by withdrawal of irrigation in the field. The findings will provide an unrivaled resource for understanding the mechanisms underlying drought resistance in pear.

## MATERIAL AND METHODS

### Plant growth conditions and drought treatment

Field drought experiments were performed for three continuous years in a pear germplasm nursery at the Institute of Fruit, Shanxi Academy of Agricultural Science, beginning on 21 October 2015 and concluding on 21 October 2018. The pear nursery is located in a semi-arid area of Taigu, Shanxi Province, China (37°26′N, 37°26′E) with an altitude of 750 m and managed according to common cultural practices in the region. In this region, the annual average temperature is 9.8 °C with an annual accumulated temperature above 10 °C (AT10) of 3529 °C. The annual hours of sunshine range from 2,500 h to 2,600 h with an average frost-free period of 149 days. The annual rainfall is 450 mm, and the annual accumulative evaporation is 1,800 mm, which is approximately four times higher than the average total rainfall.

The pear cultivar YuluXiangli (*Pyrus spp*) was used in the experiment. YuluXiangli was derived from a cross between *Pyrus bretschneiderie* and *Pyrus sinkiangensis*, and is resistant to drought. The irrigation (control) and field-drought treatments were assigned *via* a randomized block design with three replicates, where the fields were divided into six plots with 10 healthy and uniform 15-year-old pear trees per plot. Field drought plots were exposed to rainfall without additional irrigation, whereas control plots were irrigated in November, May, and July, each of which received 728.5 tons water/acre. The maximum water holding capacity was 30% in field-drought treatment (severe drought) and 75% to 80% in control with irrigation. Fertilization and pest controls were consistent among the field-drought and control plots.

In total, 100 young leaves, branches, and young fruits, including 10 from each tree, were independently harvested on 5 May 2018, and were swiftly placed in liquid nitrogen and stored at −80 °C for RNA extraction.

At maturity, 10 fruits, each from a single tree, were independently harvested to determine fruit soluble solids content with a handheld PAL-1 digital display sugar meter (Atago, Tokyo, Japan) and single fruit weight.

## Total RNA extraction, library preparation and sequencing

Total RNA was extracted from young leaves, branches, and young fruits for each treatment using RNApreo Pure Plant Kit (Tiangen, Beijing, China) in accordance with the manufacturer's instructions. RNA purity and integrity were determined by Agilent 2100 Bioanalyzer (Agilent Technology, USA) according to the manufacturer's instructions. The qualified RNA with an RNA integrity number (RIN) of $\geq 7$ and an 28S/18S ribosomal RNA ratio of $\geq 0.7$ was applied to construct 10 cDNA libraries (5 repeats for drought and irrigation, respectively). Equal amounts of RNA from young leaves, branches, and young fruits for each treatment were mixed, and then were diluted to 1 ng/$\mu$L for library construction. Briefly, RNA was enriched by magnetic beads containing poly-T oligos and fragmented first to 200–300 bp in length by ion interruption, and reversed transcribed to the first strand of cDNA by 6-bp random primers. Then, the first strand of cDNA was used as a template to synthesize the second strand of cDNA. Library fragments were enriched by PCR amplification to select the fragment size of 300–400 bp. Equal amounts of libraries with different index sequences were pooled prior to sequencing and diluted to 2 nM for paired-end sequencing on the Illumina HiSeq 2500 platform. All raw reads were deposited in the NCBI repository with Bioproject: PRJNA655255 under the accession numbers of SRR12424088–SRR12424107.

## Read mapping and transcript profiling

The adapter and low-quality sequences were removed from the raw RNA-seq reads to generate high-quality clean reads that were aligned to the pear genome reference GCF_000315295.1_Pbr_v1.0_(https://ftp.ncbi.nlm.nih.gov/genomes/all/000/315/295/GCF_000315295.1_Pbr_v1.0/) with HISAT2 (http://ccb.jhu.edu/software/hisat2/index.shtml). Following the alignments, the raw counts for each pear gene were normalized as fragments per kilobase of transcript per million mapped reads (FPKM) (*Trapnell et al., 2010*). Principal component analysis (PCA) was performed to compare the log2-transformed FPKM values of the expressed gene profiles among tissue-type and stages using the prcomp function in the R program (https://www.r-project.org/). The hierarchical clustering of samples was performed using Pheatmap in R. Read coverage over gene body was calculated by RSeQC (*Wang, Wang & Li, 2012*), and the corresponding plot figure was made by using ggplot2 with R script.

## Identification of differentially expressed genes (DEGs)

DEGs among tissue-types at different stages were located using the statistical package DEGseq with the MA-plot-based method (*Wang et al., 2010*) in R version 3.0.3, where genes were considered differentially expressed if |log2FoldChange|>1, and an adjusted *p* value using Benjamini–Hochberg procedure (*Benjamini & Hochberg, 1995*) (false discovery rate (FDR)) was <0.05.

## Gene annotation (GO) and functional enrichment analysis

The GO enrichment analysis for biological processes, molecular functions, and cellular components was performed using TopGo (*Alexa & Rahnenfuhrer, 2016*) with *P* value <0.05. Pathway enrichment analysis was implemented on all DEGs in the Kyoto Encyclopedia of Genes and Genome (KEGG) platform (http://www.genome.jp/kegg/) (*Kanehisa et al., 2008*). An adjusted *P* value <0.05 was considered statistically significant.

## Statistical analysis

Single fruit weight and soluble solid content were expressed as the mean ± standard error from 10 independent biological replicates by SPSS (V24.0, IBM Corporation, Armonk, NY, USA). These were subjected to one-way analysis of variance (ANOVA), followed by Duncan's Multiple Range post-hoc test, and the significance level was set to $P < 0.01$.

## Validation of transcripts by quantitative real-time PCR (qRT-PCR)

The expression levels of a set of randomly selected 13 DEGs were validated by a qRT-PCR assay. Total RNA used for RNA-seq was treated with RNase-free DNase I (New England Biolabs, Ipswich, MA, USA) to eradicate all contaminating DNA. A total of 1,000 ng RNA was used for the reverse transcription with PrimeScript$^{TM}$1st stand cDNA Synthesis Kit. qRT-PCR was performed with SYBR Premix Ex Taq (TaKaRa, Dalian, China) on ABI Step One RT-PCR system, according to the manufacturer's instructions (20 μL reaction mix: 1 μL cDNA, 10 $\mu$L 2×SYBR real-time PCR premixture, 0.4 μL each 10 μM primer, and 8.2 μL distilled water). Three biological replicates with two technical replicates were performed for each sample. The gene IDs and sequences of 13 primers are listed in Table 1. The PCR program was as follows: 95 °C for 5 min, followed by 40 cycles of 95 °C for 15s, and 60 °C for 30s. Relative expression was normalized to the internal control gene GAPDH gene with $2^{-\Delta\Delta CT}$ method (*Livak & Schmittgen, 2001*). Pearson's correlation was performed using R software (ver. 3.2.4, *R Core Team, 2014*) to determine the correlation of gene expression between qRT-PCR and transcriptomic data.

# RESULTS

## Effect of drought stress on physiological traits and antioxidant activities

Two irrigation treatments were applied to pear trees over the course of three continuous years. Irrigated pear trees were well irrigated, whereas pear trees subjected to deficit irrigation were not irrigated over the same period of time. As shown in Fig. 1, rainfalls during the 2018 season were extremely scarce (Fig. 1A), the consequence of which was a severe decrease in single fruit weight and soluble solids content (Figs. 1B and 1C).

## RNA-seq and de novo assembly

Paired-end RNA-Seq was performed on 10 cDNA libraries (5 repeats for drought and irrigation). Each sample was independently aligned, processed for quality control, and then normalized. A total of 400,755,040 clean reads (Table S1) were generated, among which more than 71.7% were mapped to the pear genome GCF_000315295.1_Pbr_v1.0_genomic.fna (Table 2). As indicated by FPKM, the expression

**Table 1  The gene IDs and primer sequences for qRT-PCR.**

| ID | Primer | 5′ to 3′ |
| --- | --- | --- |
| gene40303 | gene40303-F | TGGAGGCAGATAGGGTGA |
|  | gene40303-R | CCGTGTAGGAAGCAGTCG |
| gene10948 | gene10948-F | AGCCTTGCTTCTTATTCGTC |
|  | gene10948-R | ATTGCTTGAGTCCTTGCC |
| gene1490 | gene1490-F | GTGCGATTACGAGCAAGAG |
|  | gene1490-R | GAGGGGATGAAGGGTTGT |
| gene2348 | gene2348-F | GAAACCTTCACTGCCAATCT |
|  | gene2348-R | CTCATACCATCA ACCAACGA |
| gene37199 | gene37199-F | GCTTGGGTGGCGTAGTAG |
|  | gene37199-R | TCCTCCGTAATCAGGTTCTC |
| gene8748 | gene8748-F | ATGCGGATGAGCTGTAATG |
|  | gene8748-R | AGAACTTGGCGAGGAAAAC |
| gene4671 | gene4671-F | TGGACAAGAAGAAGGCAAC |
|  | gene4671-R | ATGGAAGTAAATGGCGTGA |
| gene10009 | gene10009-F | GAGATGTGAGGAGGGGAAC |
|  | gene10009-R | ATTCAGCCAGAGAGGCAA |
| gene7767 | gene7767-F | GCTGGTTGCTATGCTGGT |
|  | gene7767-R | TGTCAAGGTGGGTGTCAGT |
| gene39889 | gene39889-F | GAGATGTGAGGAGGGGAAC |
|  | gene39889-R | ATTCAGCCAGAGAGGCAA |
| gene7760 | gene7760-F | TCGTTGGTGGAAATGTTGT |
|  | gene7760-R | CAGTTGTGGTTTTGCCTTC |
| gene7261 | gene7261-F | CGATACAAGAGATGGGGAAG |
|  | gene7261-R | AGTCGGATTCACAGAAGCA |
| gene16914 | gene16914-2F | TTATTCGTTGATTCGGAACTACCA |
|  | gene16914-2R | TCTACCTCCTCCTCCTCCTT |

values showed high correlations (Spearman correlation coefficient (SCC) = 0.99) among biological replicates, which in turn demonstrated that the sequencing quality was satisfactory for subsequent analyses. Principal component analyses (PCA) revealed that the five replicates of each treatment were located nearest to each other (Fig. 2), thereby demonstrating the reliability of our datasets.

## Identification of DEGs between field drought and irrigation treatment
In total, 819 DEGs between drought and irrigation were identified by pairwise sample comparisons (Fig. 3A), among which 427 DEGs were upregulated and 392 DEGs were downregulated in comparison to that of irrigation (Table S2). The expression changes of genes in response to field drought are shown in Fig. 3B. The highly expressed (log2FoldChange $<-3.5$) drought specific genes (Fig. 3C) included gene38569 encoding Probable WRKY TF40, gene 1490 encoding WRKY TF 18, gene 30473 encoding ferritin-4, gene 7768 and gene 6357 encoding 4-hydroxycoumarin synthase 1, gene 5151 encoding

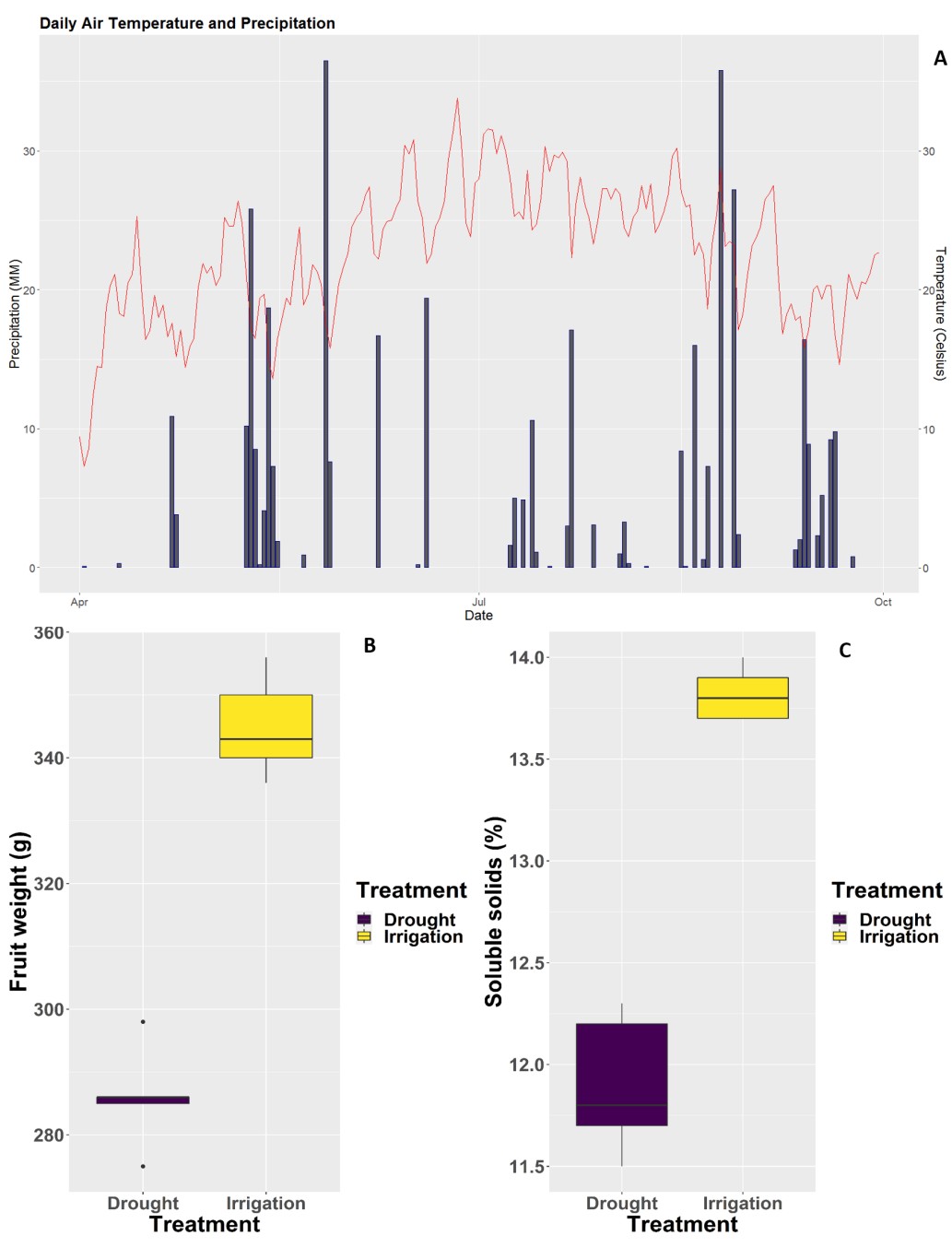

**Figure 1** **Weather conditions at the experimental site and impact of irrigation treatments on fruit weight and physiology.** (A) Daily rainfall and average temperature during the 2018 pear growth season. (B) Single fruit weight. (C) Soluble solids content.

histidine-containing phosphotransfer protein 4, gene 16914 encoding protein NIM1-INTERACTING 1, and gene12366 encoding uncharacterized protein LOC103951864 (Table 3). Genes that were highly expressed in irrigated samples but identified in drought samples included gene 27148 encoding GDL79_ARATH GDSL esterase/lipase At5g33370,

**Table 2** Summary of read numbers based on the RNA-Seq data from field drought and irrigation samples.

| Sample | Clean_Reads | Total_Mapped | Multiple_Mapped | Uniquely_Mapped |
|---|---|---|---|---|
| Drought_1 | 44498956 | 32352950(72.70%) | 3292204(10.18%) | 29060746(89.82%) |
| Drought_2 | 40138692 | 29058326(72.39%) | 2929579(10.08%) | 26128747(89.92%) |
| Drought_3 | 40161076 | 27616611(68.76%) | 3020158(10.94%) | 24596453(89.06%) |
| Drought_4 | 44937140 | 32560917(72.46%) | 3313185(10.18%) | 29247732(89.82%) |
| Drought_5 | 41366108 | 29276201(70.77%) | 3178099(10.86%) | 26098102(89.14%) |
| Irrigation_l | 40866274 | 29458607(72.09%) | 3015133(10.24%) | 26443474(89.76%) |
| Irrigation_2 | 38421602 | 28293159(73.64%) | 2752309(9.73%) | 25540850(90.27%) |
| Irrigation_3 | 36898382 | 26838597(72.74%) | 2709184(10.09%) | 24129413(89.91%) |
| Irrigation_4 | 37725898 | 26256731(69.60%) | 2917913(11.11%) | 23338818(88.89%) |
| Irrigation_5 | 35740912 | 25796677(72.18%) | 2616670(10.14%) | 23180007(89.86%) |

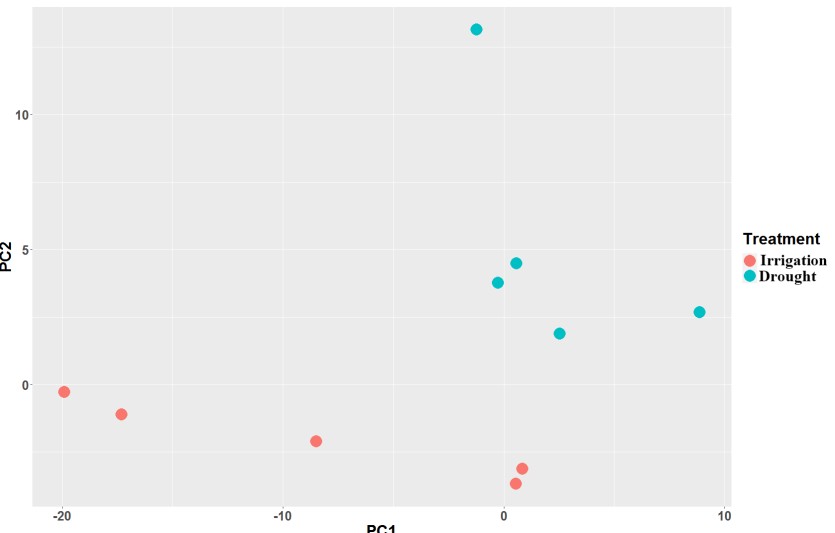

**Figure 2** Principal component analysis (PCA) of the pear transcriptome of 10 samples collected from field drought and irrigation pear trees.

gene 5286 encoding uncharacterized protein LOC103944059 isoform X1, gene 1170 encoding putative receptor protein kinase ZmPK1, gene13865 encoding gibberellin-regulated protein 11, gene19880 encoding type I inositol 1,4,5-trisphosphate 5-phosphatase CVP2-like isoform X2, and three genes (gene33465, gene39363, and gene34550) encoding palmitoyl-monogalactosyldiacylglycerol delta-7 desaturase (Fig. 3D, Table 4). The specific expression of 2 DEGs, WRKY TF 18 (gene 1490) and NIM1-INTERACTING 1 (gene 16914), was analyzed by RT-qPCR. Consistent with our RNA-seq results, WRKY TF 18 (gene 1490) was highly expressed in drought treatment at a relatively stable expression level, and the transcription of NIM1-INTERACTING 1 (gene 16914) was consistent with the RNA-seq result only in the irrigation (Fig. 4).

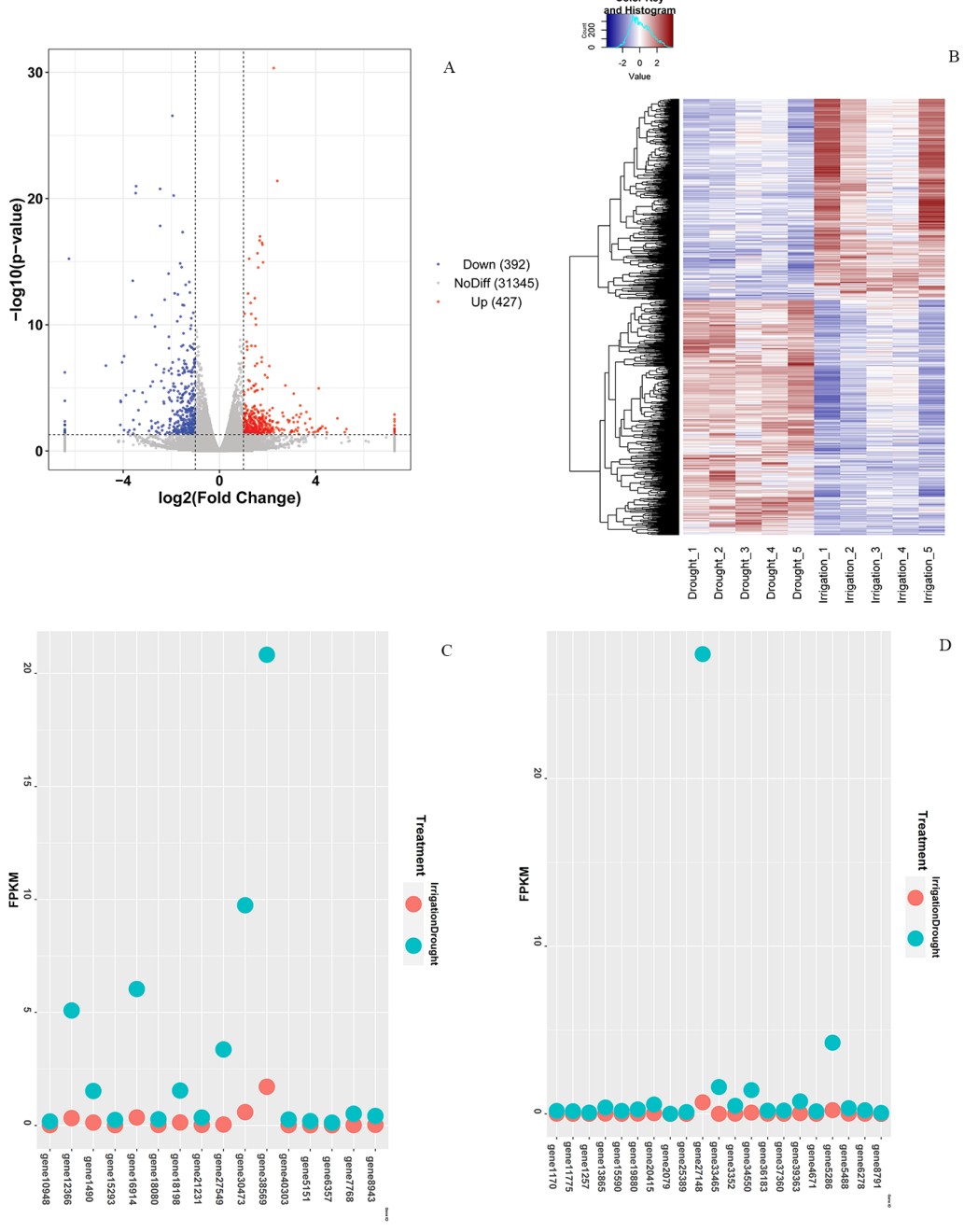

**Figure 3 Differential expression analysis.** (A) Venn diagram of DEGs between drought and irrigation treatment. (B) Heat map of the DEG expression levels. (C) Highly expressed genes (log2FoldChange < −3.5) exclusively identified in field drought samples. (D) Highly expressed genes identified exclusively in irrigation samples.

## Co-expression analysis of DEGs during field drought treatment

In order to investigate the co-expressed genes during field drought stress, all the genes that were differentially expressed between drought and irrigation were statistically clustered into

**Table 3  Highly expressed genes identified in samples under field drought conditions (log2FoldChange <−3.5).**

| Gene_ID | Irrigation_1.fpkm | Irrigation_2.fpkm | Irrigation_3.fpkm | Irrigation_4.fpkm | Irrigation_5.fpkm | Drought_1.fpkm | Drought_2.fpkm | Drought_3.fpkm | Drought_4.fpkm | Drought_5.fpkm | Swissprot | Description |
|---|---|---|---|---|---|---|---|---|---|---|---|---|
| gene 10948 | 0.16 | 0 | 0.06 | 0.26 | 0.41 | 0 | 0 | 0 | 0.07 | 0 | LOR6_ARATH Protein LURP-one-related 6 OS=Arabidopsis thaliana GN=At2g05910 PE=2 SV=1 | protein LURP-one-related 6-like [Pyrus x bretschneideri] |
| gene 12366 | 2.59 | 3.46 | 6.12 | 9.01 | 4.27 | 0.14 | 0.22 | 0.7 | 0.24 | 0.32 | | uncharacterized protein LOC103951864 [Pyrus x bretschneideri] |
| gene 1490 | 0.52 | 1.22 | 1.03 | 2.73 | 2.14 | 0.33 | 0 | 0.05 | 0.16 | 0.11 | WRK40_ARATH Probable WRKY transcription factor 40 OS=Arabidopsis thaliana GN=WRKY40 PE=1 SV=1 | WRKY transcription factor 18 [Pyrus x bretschneideri] |
| gene 15293 | 0.18 | 0.17 | 0.39 | 0.36 | 0.13 | 0 | 0.03 | 0 | 0.04 | 0 | ACA12_ARATH Calcium-transporting ATPase 12, plasma membrane-type OS=Arabidopsis thaliana GN=ACA12 PE=2 SV=1 | calcium-transporting ATPase 12, plasma membrane-type-like [Pyrus x bretschneideri] |
| gene 16914 | 2.37 | 3.21 | 12.8 | 6.9 | 4.92 | 0.74 | 0.19 | 0 | 0.85 | 0 | | protein NIM1-INTERACTING 1 [Pyrus x bretschneideri] |
| gene 18080 | 0.55 | 0.22 | 0.23 | 0.2 | 0.17 | 0.05 | 0 | 0.06 | 0 | 0 | RKD4_ARATH Protein RKD4 OS=Arabidopsis thaliana GN=RKD4 PE=3 SV=1 | uncharacterized protein LOC103948099 [Pyrus x bretschneideri] |
| gene 18198 | 0.42 | 0.61 | 3.77 | 2.83 | 0.09 | 0 | 0 | 0.45 | 0.11 | 0.11 | YE04_SCHPO Uncharacterized RNA-binding protein C17H9.04c OS=Schizosaccharomyces pombe (strain 972/ATCC 24843) GN=SPAC17H9.04c PE=1 SV=1 | uncharacterized RNA-binding protein C17H9.04c [Pyrus x bretschneideri] |
| gene 21231 | 0 | 0.12 | 0.86 | 0.31 | 0.46 | 0 | 0 | 0.12 | 0 | 0 | | uncharacterized protein LOC103961606 [Pyrus x bretschneideri] |
| gene 27549 | 3.17 | 3.81 | 2.73 | 1.63 | 5.48 | 0 | 0.14 | 0.04 | 0.04 | 0 | REXO4_YEAST RNA exonuclease 4 OS=Saccharomyces cerevisiae (strain ATCC 204508/S288c) GN=REX4 PE=1 SV=1 | RNA exonuclease 4 [Pyrus x bretschneideri] |
| gene 30473 | 4.38 | 5.38 | 13.7 | 9.03 | 16.2 | 0 | 0.13 | 1.09 | 0.66 | 1.09 | FRI3_SOYBN Ferritin-3, chloroplastic OS=Glycine max PE=2 SV=1 | ferritin-4, chloroplastic-like [Pyrus x bretschneideri] |
| gene 38569 | 12.6 | 14.1 | 22.1 | 27 | 28.3 | 2.01 | 1.2 | 3.31 | 1.66 | 0.4 | WRK40_ARATH Probable WRKY transcription factor 40 OS=Arabidopsis thaliana GN=WRKY40 PE=1 SV=1 | probable WRKY transcription factor 40 [Pyrus x bretschneideri] |
| gene 40303 | 0.31 | 0.07 | 0.07 | 0.25 | 0.62 | 0 | 0 | 0 | 0 | 0.08 | | uncharacterized protein LOC103940893 [Pyrus x bretschneideri] |
| gene 5151 | 0.49 | 0 | 0.13 | 0.22 | 0.12 | 0 | 0.06 | 0 | 0 | 0 | AHP4_ARATH Histidine-containing phosphotransfer protein 4 OS=Arabidopsis thaliana GN=AHP4 PE=1 SV=2 | histidine-containing phosphotransfer protein 4-like [Pyrus x bretschneideri] |
| gene 6357 | 0.25 | 0 | 0.15 | 0.17 | 0.05 | 0 | 0 | 0.05 | 0 | 0 | BIPS2_SORAU 4-hydroxycoumarin synthase 1 OS=Sorbus aucuparia GN=BIS2 PE=1 SV=1 | 4-hydroxycoumarin synthase 1-like [Pyrus x bretschneideri] |

| Gene_ID | Irrigation_1.fpkm | Irrigation_2.fpkm | Irrigation_3.fpkm | Irrigation_4.fpkm | Irrigation_5.fpkm | Drought_1.fpkm | Drought_2.fpkm | Drought_3.fpkm | Drought_4.fpkm | Drought_5.fpkm | Swissprot | Description |
|---|---|---|---|---|---|---|---|---|---|---|---|---|
| gene 7768 | 0.49 | 0.45 | 0.53 | 0.64 | 0.5 | 0 | 0 | 0.1 | 0 | 0 | BIPS2_SORAU 4-hydroxycoumarin synthase 1 OS=Sorbus aucuparia GN=BIS2 PE=1 SV=1 | 4-hydroxycoumarin synthase 1 [Pyrus x bretschneideri] |
| gene 8943 | 0.56 | 0.53 | 0.25 | 0.34 | 0.43 | 0.05 | 0.1 | 0 | 0 | 0 | RKD4_ARATH Protein RKD4 OS=Arabidopsis thaliana GN=RKD4 PE=3 SV=1 | uncharacterized protein LOC103948099 [Pyrus x bretschneideri] |

different groups according to their expression profiles. The largest group (Fig. 5A) included 539 genes that predominantly annotated to RLP12_ARATH and increasingly expressed under field drought conditions. Receptor-like protein 12 participated in the perception of CLV3 and CLV3-like peptides to act as extracellular signals regulating meristems maintenance (149/539). ZIFL1_ARATH Protein ZINC INDUCED FACILITATOR-LIKE 1 (120/539), TMVRN_NICGU TMV resistance protein N (90/539), Y3475_ARATH Probable LRR receptor-like serine/threonine-protein kinase At3g47570 (86/539), and WRK40_ARATH Probable WRKY transcription factor 40 were responsible for the regulation of genes responsive to biotic and abiotic stresses (79/539). The second largest group (Fig. 5B) contained 293 genes whose expression increased under field drought conditions. Genes in this cluster were mainly annotated to BAMS_BETPL Beta-amyrin synthase, which catalyzes the formation of the most popular triterpene among higher plants, HDAC6_HUMAN Histone deacetylase 6, HDAC6_HUMAN Histone deacetylase 6, KAP1_ARATH Adenylyl-sulfate kinase 1, chloroplastic, and RAP24_ARATH Ethylene-responsive transcription factor RAP2-4. The third largest group (Fig. 5C) contained 35 genes whose expression decreased under field drought conditions.

## Functional analysis of DEGs between drought and irrigation

Functional analysis was performed to locate enriched GO terms and KEGG pathways involving the DEGs. As shown in Table 5, DEGs were significantly assigned to microtubule (GO:0005874), polymeric cytoskeletal fiber (GO:0099513), and supramolecular complex (GO:0099080) in the cell component (CC) category. In the molecular function (MF) category, DEGs were primarily assigned to microtubules motor activity (GO:0003777), motor activity (GO:0003774), and microtubules binding (GO:0008017). In the biological process (BP) category, DEGs were mainly assigned to microtubules-based movement (GO:0007018) and the movement of cell or subcellular components (GO:0006928). These results demonstrate that DEGs involved in binding, transport, and movement were critical during drought stress.

KEGG pathway enrichment analysis revealed that DEGs were notably enriched in plant monoterpenoid biosynthesis, flavonoid biosynthesis, diterpenoid biosynthesis, cysteine and methionine metabolism, phenylpropanoid biosynthesis, and carotenoid biosynthesis (Fig. 6, Table S3), suggesting specific metabolic events during drought. DEGs were identified using the log2 fold change of the transcript level in field drought compared to the irrigation, and were mapped into the related metabolic pathways (Table 6), thereby revealing a significant impact of field drought on secondary metabolism. Field

**Table 4 Highly expressed genes identified in samples under irrigation conditions (log2FoldChange < −3.5).**

| Gene_ID | Irrigation_1.fpkm | Irrigation_2.fpkm | Irrigation_3.fpkm | Irrigation_4.fpkm | Irrigation_5.fpkm | Drought_1.fpkm | Drought_2.fpkm | Drought_3.fpkm | Drought_4.fpkm | Drought_5.fpkm | Swissprot | Description |
|---|---|---|---|---|---|---|---|---|---|---|---|---|
| gene 1170 | 0.03 | 0 | 0 | 0 | 0.04 | 0.11 | 0.22 | 0.24 | 0.37 | 0 | KPRO_MAIZE Putative receptor protein kinase ZmPK1 OS=Zea mays GN=PK1 PE=2 SV=2 | putative receptor protein kinase ZmPK1 [Pyrus x bretschneideri] |
| gene 11775 | 0.02 | 0 | 0 | 0 | 0 | 0.17 | 0.08 | 0.03 | 0.03 | 0.53 | GDL82_ARATH GDSL esterase/lipase At5g45670 OS=Arabidopsis thaliana GN=At5g45670 PE=2 SV=1 | GDSL esterase/lipase At5g45670-like [Malus domestica] |
| gene 1257 | 0.01 | 0 | 0 | 0.03 | 0 | 0.19 | 0.08 | 0.01 | 0.02 | 0.13 | NACK1_TOBAC Kinesin-like protein NACK1 OS=Nicotiana tabacum GN=NACK1 PE=1 SV=1 | uncharacterized protein LOC103955247 [Pyrus x bretschneideri] |
| gene 13865 | 0 | 0 | 0 | 0.09 | 0 | 0.2 | 1.05 | 0.11 | 0.35 | 0.23 | SNAK2_SOLTU Snakin-2 OS=Solanum tuberosum GN=SN2 PE=1 SV=1 | gibberellin-regulated protein 11-like [Pyrus x bretschneideri] |
| gene 15590 | 0 | 0 | 0 | 0.05 | 0 | 0.17 | 0.06 | 0.06 | 0.25 | 0.38 | RADL1_ARATH Protein RADIALIS-like 1 OS=Arabidopsis thaliana GN=RL1 PE=2 SV=1 | protein RADIALIS-like 3 [Malus domestica] gi\|694378665\|ref\| XP_009365559.1\| PR |
| gene 19880 | 0 | 0.03 | 0 | 0.05 | 0 | 0.44 | 0.14 | 0.15 | 0.19 | 0.44 | IP5P3_ARATH Type I inositol 1,4,5-trisphosphate 5-phosphatase CVP2 OS=Arabidopsis thaliana GN=CVP2 PE=1 SV=2 | type I inositol 1,4,5-trisphosphate 5-phosphatase CVP2-like isoform X2 [Pyrus |
| gene 19880 | 0 | 0.03 | 0 | 0.05 | 0 | 0.44 | 0.14 | 0.15 | 0.19 | 0.44 | IP5P3_ARATH Type I inositol 1,4,5-trisphosphate 5-phosphatase CVP2 OS=Arabidopsis thaliana GN=CVP2 PE=1 SV=2 | type I inositol 1,4,5-trisphosphate 5-phosphatase CVP2-like isoform X1 [Pyrus |
| gene 20415 | 0 | 0 | 0 | 0.23 | 0 | 0.17 | 0.53 | 0.47 | 0.78 | 0.88 |  | transcription repressor OFP8-like [Pyrus x bretschneideri] |
| gene 2079 | 0 | 0 | 0 | 0 | 0 | 0.03 | 0.01 | 0.01 | 0 | 0.02 |  |  |
| gene 25389 | 0.03 | 0 | 0 | 0 | 0 | 0.26 | 0.03 | 0.03 | 0 | 0.26 | AB8G_ARATH ABC transporter G family member 8 OS=Arabidopsis thaliana GN=ABCG8 PE=2 SV=1 | ABC transporter G family member 4-like [Pyrus x bretschneideri] gi\|694405461\| |
| gene27148 | 2.16 | 0.19 | 0.69 | 0.24 | 0.19 | 43.7 | 6.64 | 0.15 | 0.21 | 86.4 | GDL79_ARATH GDSL esterase/lipase At5g33370 OS=Arabidopsis thaliana GN=At5g33370 PE=2 SV=1 | GDSL esterase/lipase At5g33370-like [Pyrus x bretschneideri] |
| gene 33465 | 0 | 0 | 0.05 | 0 | 0 | 1.89 | 0.57 | 0.41 | 0.42 | 4.75 | ADS3_ARATH Palmitoyl-monogalactosyldiacylglycerol delta-7 desaturase, chloroplastic OS=Arabidopsis thaliana GN=ADS3 PE=2 SV=2 | palmitoyl-monogalactosyldiacylglycerol delta-7 desaturase, chloroplastic-like |
| gene 3352 | 0.04 | 0 | 0 | 0 | 0.04 | 1.25 | 0.47 | 0.05 | 0.19 | 0.48 | NAC98_ARATH Protein CUP-SHAPED COTYLEDON 2 OS=Arabidopsis thaliana GN=NAC098 PE=1 SV=1 | protein CUP-SHAPED COTYLEDON 2 [Pyrus x bretschneideri] |
| gene 34550 | 0 | 0.09 | 0.23 | 0 | 0.09 | 1.79 | 0.75 | 0 | 0.19 | 4.39 | ADS3_ARATH Palmitoyl-monogalactosyldiacylglycerol delta-7 desaturase, chloroplastic OS=Arabidopsis thaliana GN=ADS3 PE=2 SV=2 | palmitoyl-monogalactosyldiacylglycerol delta-7 desaturase, chloroplastic-like |

**Table 4** (*continued*)

| Gene_ID | Irrigation _1.fpkm | Irrigation _2.fpkm | Irrigation _3.fpkm | Irrigation _4.fpkm | Irrigation _5.fpkm | Drought _1.fpkm | Drought _2.fpkm | Drought _3.fpkm | Drought _4.fpkm | Drought _5.fpkm | Swissprot | Description |
|---|---|---|---|---|---|---|---|---|---|---|---|---|
| gene 36183 | 0 | 0 | 0 | 0.08 | 0 | 0.43 | 0 | 0.1 | 0.1 | 0.35 | IQD31_ARATH Protein IQ-DOMAIN 31 OS=Arabidopsis thaliana GN=IQD31 PE=1 SV=1 | uncharacterized protein LOC103443739 [Malus domestica] gi\|657977866\|ref\|XP_00 |
| gene 37360 | 0 | 0.08 | 0 | 0 | 0 | 0.46 | 0.32 | 0.17 | 0 | 0.09 | | uncharacterized protein LOC103937664 [Pyrus x bretschneideri] |
| gene 39363 | 0.09 | 0.05 | 0 | 0 | 0.1 | 0.77 | 0.1 | 0.16 | 0.44 | 2.3 | ADS3_ARATH Palmitoyl-monogalactosyldiacylglycerol delta-7 desaturase, chloroplastic OS=Arabidopsis thaliana GN=ADS3 PE=2 SV=2 | palmitoyl-monogalactosyldiacylglycerol delta-7 desaturase, chloroplastic-like |
| gene 39363 | 0.09 | 0.05 | 0 | 0 | 0.1 | 0.77 | 0.1 | 0.16 | 0.44 | 2.3 | ADS3_ARATH Palmitoyl-monogalactosyldiacylglycerol delta-7 desaturase, chloroplastic OS=Arabidopsis thaliana GN=ADS3 PE=2 SV=2 | palmitoyl-monogalactosyldiacylglycerol delta-7 desaturase, chloroplastic-like |
| gene 4671 | 0 | 0 | 0 | 0.05 | 0 | 0.1 | 0.05 | 0.11 | 0.12 | 0.42 | | uncharacterized protein LOC103943512 [Pyrus x bretschneideri] |
| gene 5286 | 0.25 | 0.09 | 0.39 | 0.41 | 0 | 6.9 | 1.95 | 0.2 | 0.51 | 11.7 | | uncharacterized protein LOC103944059 isoform X1 [Pyrus x bretschneideri] |
| gene 5286 | 0.25 | 0.09 | 0.39 | 0.41 | 0 | 6.9 | 1.95 | 0.2 | 0.51 | 11.7 | | uncharacterized protein LOC103944059 isoform X2 [Pyrus x bretschneideri] |
| gene 5488 | 0.05 | 0.03 | 0 | 0.02 | 0 | 0.84 | 0.11 | 0.06 | 0.03 | 0.72 | GRF4_ARATH Growth-regulating factor 4 OS=Arabidopsis thaliana GN=GRF4 PE=2 SV=1 | growth-regulating factor 3-like isoform X1 [Pyrus x bretschneideri] |
| gene 6278 | 0 | 0.04 | 0 | 0 | 0.04 | 0.22 | 0.08 | 0.12 | 0.08 | 0.59 | C79D4_LOTJA Isoleucine N-monooxygenase 2 OS=Lotus japonicus GN=CYP79D4 PE=1 SV=1 | isoleucine N-monooxygenase 2-like [Pyrus x bretschneideri] |
| gene 8791 | 0 | 0 | 0 | 0.02 | 0 | 0.02 | 0.1 | 0.04 | 0.06 | 0.11 | BGL12_ORYSJ Beta-glucosidase 12 OS=Oryza sativa subsp. japonica GN=BGLU12 PE=2 SV=2 | beta-glucosidase 12-like [Pyrus x bretschneideri] |

drought modulated the expression of many DEGs that codify for structural enzymes of the monoterpenoid biosynthesis, flavonoid pathway, and phenylpropanoid biosynthesis (Table 6); the majority of these genes were downregulated under field drought. Four DEGs including the salutaridine reductase-like (SalR) gene family (gene18404, gene39888, gene39889, and gene10010) and nerolidol synthase 1-like (gene237) were involved in monoterpenoid biosynthesis, all of which were downregulated (Table 7) in response to drought stress. Drought modulated the expression of the majority of the structural flavonoid genes (Table 7), most notably three 3,5-dihydroxybiphenyl synthase-like (gene7767, gene7762, and gene 6358), one leucoanthocyanidin reductase-like isoform X1 (gene3879), one BAHD acyltransferase *At5g47980*-like (gene7261), one salutaridinol 7-O-acetyltransferase-like (gene10701), one vinorine synthase-like (gene34704), and 4-hydroxycoumarin synthase 2 (gene7760). All the aforementioned genes were upregulated by drought. The specific expression of 4 DEGs SalR (gene39889), 3,5-dihydroxybiphenyl

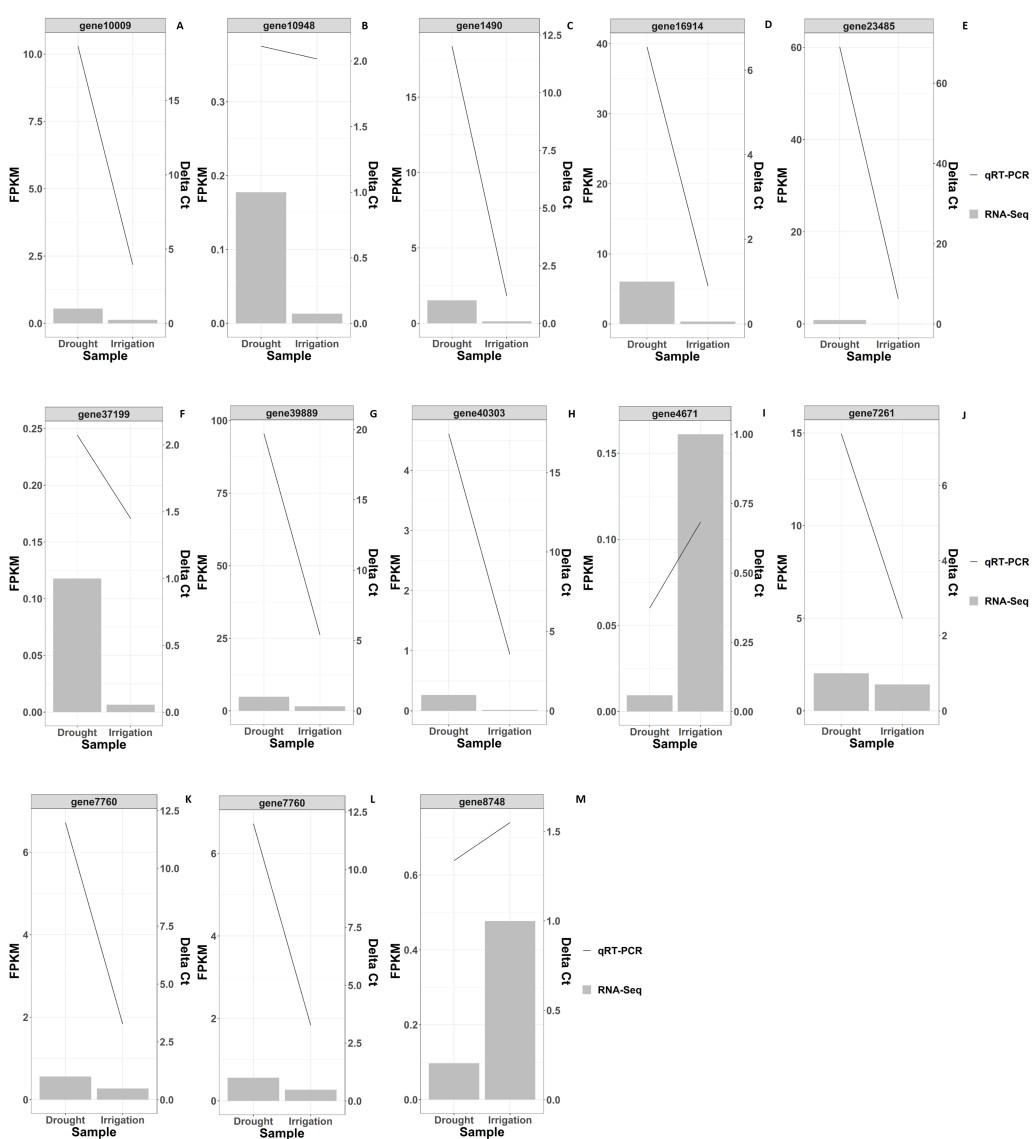

**Figure 4 RNAseq and qRT-PCR validation results of differential gene expression under drought and irrigation.** The left $Y$-axis indicates the gene expression levels calculated by the RPKM method. The right $Y$-axis indicates the relative gene expression levels.

synthase (gene7767), BAHD acyltransferase (gene7261), and 4-hydroxycoumarin synthase 2 (gene7760) was analyzed by RT-qPCR, and proved consistent with our RNA-seq results of high expression in drought treatment at a relatively stable expression level (Fig. 4).

## Differentially expressed transcription factors under drought stress

Transcription factors (TFs) play key regulatory roles in plant signaling responses, those which activate or inhibit gene expression at the transcriptional level in response to stress. Field-drought treatment led to a number of TFs being differentially expressed (Fig. 7). In total, 4438 differentially expressed TFs were identified, belonging to 30 TF families

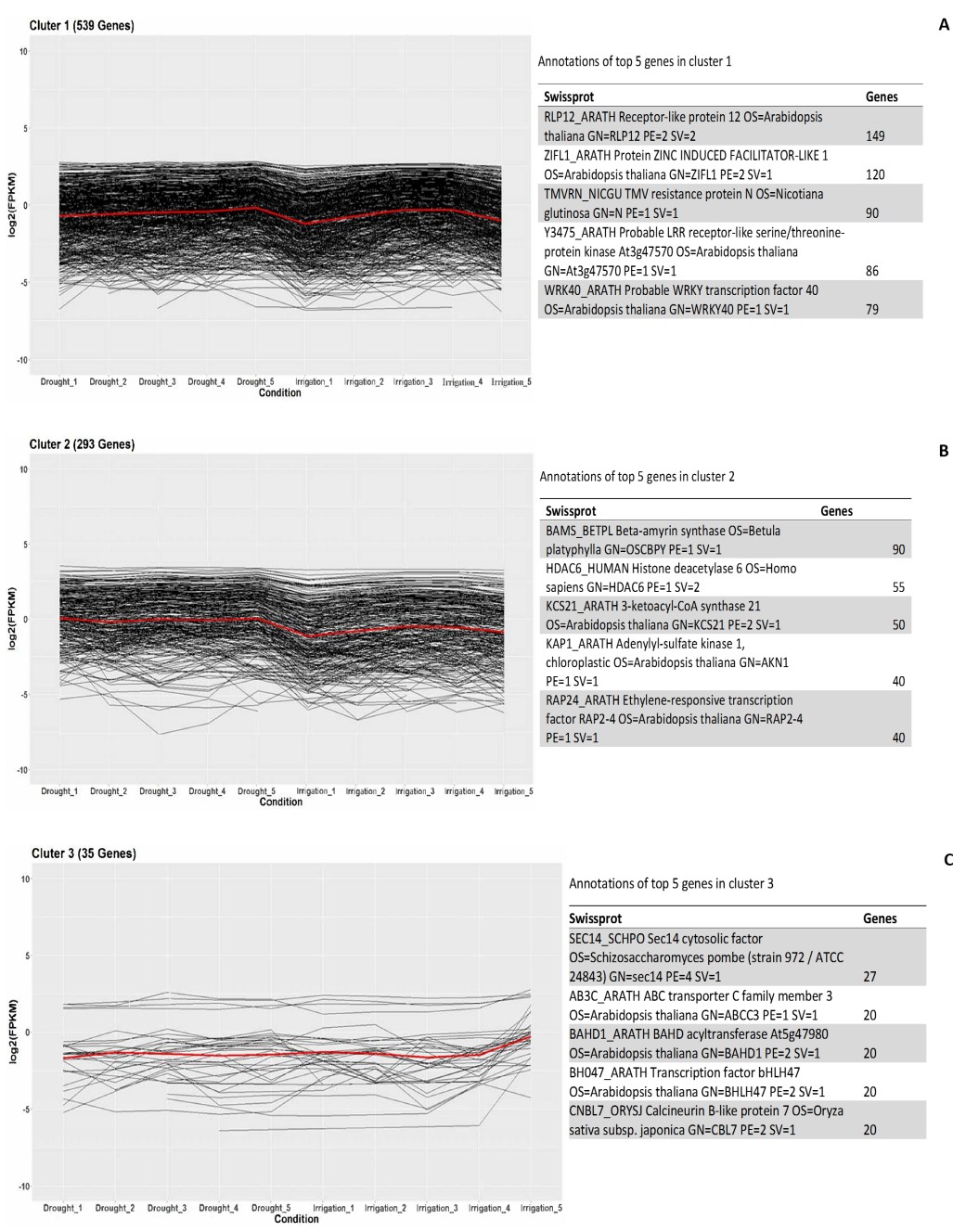

**Figure 5** Clustering and gene ontology enrichment of DEGs between drought and irrigation treatment (A–C).

such as bHLHs (basic helix-loop-helix), NAC (NAM/ATAF/CUC), MYB (v-myb avian myeloblastosis viral oncogene homolog), ERF (ethylene-responsive element binding factor), C2H2s and C3Hs (C2H2 and C3H zinc-finger proteins), WRKYs (WRKY proteins), and bZIPs (basic region-leucine zipper).

**Table 5** Top 10 GO terms of DEGs for each of the three GO categories in field drought samples compared to irrigation samples.

| Category | GO.id | Term | Up | Down | DEG | Total | *P* value | FDR |
|---|---|---|---|---|---|---|---|---|
| BP | GO:0007018 | microtubule-based movement | 14 | 0 | 14 | 44 | 7.10E−14 | 4.72E−11 |
| BP | GO:0006928 | movement of cell or subcellular component | 14 | 0 | 14 | 45 | 1.00E−13 | 4.72E−11 |
| BP | GO:0007017 | microtubule-based process | 14 | 1 | 15 | 118 | 1.20E−08 | 1.62E−06 |
| BP | GO:0007349 | cellularization | 3 | 0 | 3 | 5 | 7.60E−05 | 0.003257636 |
| BP | GO:0009558 | embryo sac cellularization | 3 | 0 | 3 | 5 | 7.60E−05 | 0.003257636 |
| BP | GO:0008150 | biological _process | 111 | 95 | 206 | 9279 | 0.00012 | 0.004715 |
| BP | GO:0019748 | secondary metabolic process | 1 | 5 | 6 | 41 | 0.00015 | 0.005440385 |
| BP | GO:0055072 | iron ion homeostasis | 1 | 2 | 3 | 7 | 0.00026 | 0.008756429 |
| BP | GO:0009698 | phenylpropanoid metabolic process | 1 | 4 | 5 | 31 | 0.00034 | 0.009926316 |
| BP | GO:0019318 | hexose metabolic process | 0 | 5 | 5 | 32 | 0.00039 | 0.009926316 |
| MF | GO:0003777 | microtubule motor activity | 14 | 0 | 14 | 44 | 3.30E−13 | 1.04E−10 |
| MF | GO:0003774 | motor activity | 14 | 0 | 14 | 45 | 4.70E−13 | 1.11E−10 |
| MF | GO:0008017 | microtubule binding | 14 | 0 | 14 | 60 | 3.60E−11 | 6.79E−09 |
| MF | GO:0015631 | tubulin binding | 14 | 0 | 14 | 73 | 5.90E−10 | 9.27E−08 |
| MF | GO:0008092 | cytoskeletal protein binding | 14 | 1 | 15 | 115 | 3.60E−08 | 2.68E−06 |
| MF | GO:0019825 | oxygen binding | 1 | 2 | 3 | 4 | 4.30E−05 | 0.00202745 |
| MF | GO:0033815 | biphenyl synthase activity | 0 | 3 | 3 | 4 | 4.30E−05 | 0.00202745 |
| MF | GO:0003824 | catalytic activity | 93 | 67 | 160 | 5794 | 0.0001 | 0.0041 |
| MF | GO:0003674 | molecular _function | 139 | 109 | 248 | 10119 | 0.00015 | 0.005440385 |
| MF | GO:0016787 | hydrolase activity | 47 | 18 | 65 | 1920 | 0.00025 | 0.008731481 |
| CC | GO:0005874 | microtubule | 11 | 0 | 11 | 70 | 2.70E−08 | 2.68E−06 |
| CC | GO:0099513 | polymeric cytoskeletal fiber | 11 | 0 | 11 | 71 | 3.20E−08 | 2.68E−06 |
| CC | GO:0099080 | supramolecular complex | 11 | 0 | 11 | 72 | 3.70E−08 | 2.68E−06 |
| CC | GO:0099081 | supramolecular polymer | 11 | 0 | 11 | 72 | 3.70E−08 | 2.68E−06 |
| CC | GO:0099512 | supramolecular fiber | 11 | 0 | 11 | 72 | 3.70E−08 | 2.68E−06 |
| CC | GO:0015630 | microtubule cytoskeleton | 11 | 1 | 12 | 107 | 3.00E−07 | 1.95E−05 |
**Table 5** (*continued*)

| Category | GO.id | Term | Up | Down | DEG | Total | *P* value | FDR |
|----------|-------|------|----|------|-----|-------|-----------|-----|
| CC | GO:0005576 | extracellular region | 11 | 7 | 18 | 250 | 3.10E−07 | 1.95E−05 |
| CC | GO:0044430 | cytoskeletal part | 11 | 1 | 12 | 124 | 1.50E−06 | 8.84E−05 |
| CC | GO:0048046 | apoplast | 9 | 1 | 10 | 89 | 2.90E−06 | 0.000160865 |
| CC | GO:0005856 | cytoskeleton | 11 | 1 | 12 | 139 | 4.90E−06 | 0.000256706 |

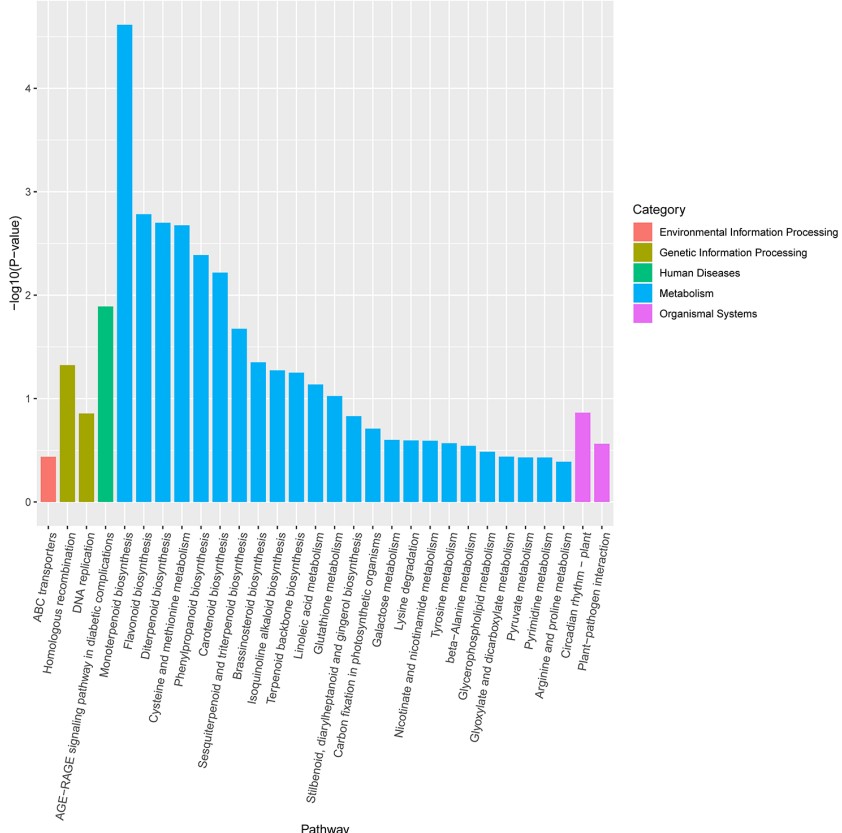

**Figure 6** **KEGG enrichment pathway analysis of differentially expressed genes between drought and irrigation pear trees.**

## Validation of DEG-based gene expression

In order to validate the RNA-Seq gene expression results, qRT-PCR was performed to evaluate the expression levels of the 13 randomly selected DEGs in irrigation *vs* field-drought conditions (Table 7). As shown in Fig. 4, the expression of the 13 DEGs was largely identical between RNA-Seq and qRT-PCR in spite of certain differences in the absolute fold change. The verified results from the qRT-PCR demonstrated trends similar to the transcriptomic results, which suggests that these DEGs could play significant roles in the regulation of production performance under field-drought conditions.

**Table 6   Top 10 pathways in metabolism related to DEGs in field drought samples compared to irrigation conditions.**

| Pathway | Level1 | Level2 | Up | Down | DEG | total _number | *P* value | FDR |
|---|---|---|---|---|---|---|---|---|
| Fatty acid elongation | Metabolism | Lipid metabolism | 3 | 4 | 7 | 60 | 0.000222794 | 0.015372772 |
| Monoterpenoid biosynthesis | Metabolism | Metabolism of terpenoids and polyketides | 0 | 4 | 4 | 21 | 0.000801546 | 0.018443153 |
| Sesquiterpenoid and triterpenoid biosynthesis | Metabolism | Metabolism of terpenoids and polyketides | 0 | 5 | 5 | 36 | 0.000801876 | 0.018443153 |
| Phenylpropanoid biosynthesis | Metabolism | Biosynthesis of other secondary metabolites | 5 | 9 | 14 | 303 | 0.004178352 | 0.057661261 |
| Carbon fixation in photosynthetic organisms | Metabolism | Energy metabolism | 1 | 6 | 7 | 104 | 0.005694659 | 0.065488575 |
| Selenocompound metabolism | Metabolism | Metabolism of other amino acids | 1 | 2 | 3 | 21 | 0.008750089 | 0.077724824 |
| Cutin, suberine and wax biosynthesis | Metabolism | Lipid metabolism | 3 | 1 | 4 | 40 | 0.009011574 | 0.077724824 |
| Flavonoid biosynthesis | Metabolism | Biosynthesis of other secondary metabolites | 0 | 5 | 5 | 75 | 0.01920126 | 0.132488694 |
| Cysteine and methionine metabolism | Metabolism | Amino acid metabolism | 5 | 2 | 7 | 154 | 0.040390415 | 0.232244887 |
| Cyanoamino acid metabolism | Metabolism | Metabolism of other amino acids | 4 | 1 | 5 | 118 | 0.09692841 | 0.514466178 |

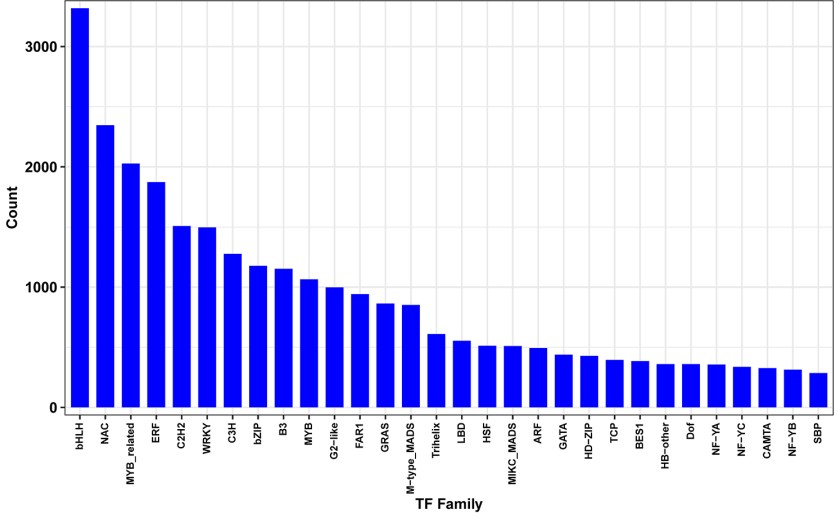

**Figure 7   Differentially expressed transcription factors genes between drought and irrigation treatment.**

## DISCUSSION

Drought is one of the vital factors limiting plant growth and distribution. Understanding the complex mechanisms of drought responses in plants is essential for improving

**Table 7 Effects of drought on monoterpenoid pathway and flavonoid biosynthesis.**

| Pathway | Gene id | foldChange | log2FoldChange | Gene prediction |
|---|---|---|---|---|
| Monoterpenoid biosynthesis | gene 18404 | 0.645958365 | −0.630486914 | Salutaridine reductase-like [Pyrus × bretschneideri] |
| Monoterpenoid biosynthesis | gene 39888 | 0.423999841 | −1.23786437 | Salutaridine reductase-like [Pyrus × bretschneideri] |
| Monoterpenoid biosynthesis | gene 39889 | 0.318598399 | −1.650189078 | Salutaridine reductase-like [Pyrus × bretschneideri] |
| Monoterpenoid biosynthesis | gene10009 | 0.233266788 | −2.099947181 | Salutaridine reductase-like isoform X1 [Pyrus × bretschneideri] |
| Monoterpenoid biosynthesis | gene10010 | 0.265228317 | −1.914693282 | Salutaridine reductase-like [Pyrus × bretschneideri] |
| Monoterpenoid biosynthesis | gene237 | 0.931868371 | −0.101801911 | (3S,6E)-nerolidol synthase 1-like [Pyrus × bretschneideri] |
| Flavonoid biosynthesis | gene7767 | −1.533981312 | 2.66172E−12 | 3,5-dihydroxybiphenyl synthase-like [Pyrus × bretschneideri] |
| Flavonoid biosynthesis | gene7762 | −0.49454945 | 0.307982886 | 3,5-dihydroxybiphenyl synthase-like [Pyrus × bretschneideri] |
| Flavonoid biosynthesis | gene3879 | -Inf | 1 | Leucoanthocyanidin reductase-like isoform X1 [Pyrus × bretschneideri] |
| Flavonoid biosynthesis | gene7261 | −0.510002646 | 0.062753961 | BAHD acyltransferase At5g47980-like [Pyrus × bretschneideri] |
| Flavonoid biosynthesis | gene10701 | −0.035249252 | 0.880552832 | Salutaridinol 7-O-acetyltransferase-like [Pyrus × bretschneideri] |
| Flavonoid biosynthesis | gene34704 | 0.004359729 | 0.96585065 | Vinorine synthase-like [Pyrus × bretschneideri] |
| Flavonoid biosynthesis | gene6358 | -Inf | 0.000106527 | 3,5-dihydroxybiphenyl synthase-like [Pyrus × bretschneideri] |
| Flavonoid biosynthesis | gene7760 | −1.047453808 | 0.04585677 | 4-hydroxycoumarin synthase 2 [Pyrus × bretschneideri] |

drought tolerance through programmed selection with precise strategies of stress-testing, particularly in light of ongoing global climate change. In the present study, we identified differentially expressed genes under field-drought stress and irrigation control with RNA-Seq in the pear cultivar YuluXiangli. A total of 819 DEGs were detected, and 4,438 TFs were differentially expressed between drought and irrigation control. Our findings represent valuable information on transcriptome changes in response to drought. Drought responsive genes are mainly enriched in biosynthesis-related pathways—monoterpenoid biosynthesis, flavonoid biosynthesis, and diterpenoid biosynthesis—and they belong mainly to bHLHs, NAC, MYB, ERF, C2H2s, and C3Hs, as well as to WRKYs transcription factor families. Our analysis provides a solid foundation for both the identification and the functional analysis of potential candidate genes related to drought tolerance.

The prolonged and severe field drought imposed in this experiment modulated the accumulation of phenylpropanoids, flavonoids, monoterpenoid biosynthesis, and several volatile organic compounds in the pear. Previous studies demonstrated the drought-modulated accumulation of phenylpropanoids, flavonoids, terpenoids, and carotenoids under drought (*Li et al., 2018*; *Murphy & Zerbe, 2020*; *Savoi et al., 2016*; *Sircelj et al., 2005*). This accumulation acted as antioxidants and protected plants from the adverse effects of drought conditions (*Nichols, Hofmann & Williams, 2015*). Our study demonstrated

modulation of the biosynthetic pathways of phenylpropanoids and flavonoids by drought stress at the transcript level, leading to an enhanced accumulation of derivatives of benzoic and cinnamic acids as well as several flavonoids. This was congruent with previous results (*Li et al., 2018*; *Murphy & Zerbe, 2020*; *Nichols, Hofmann & Williams, 2015*; *Savoi et al., 2016*; *Sircelj et al., 2005*). Five of 14 phenylpropanoid DEGs as well as all of the flavonoid DEGs were upregulated under drought stress, the result of which enhanced the concentration of accumulation within these compounds. Flavonoid aggregation in cytoplasm is capable of effectively detoxifying drought-induced harmful $H_2O_2$ molecules. In the present study, the elevated flavonoid aggregation was induced by drought stress condition, supporting previous results in *Achillea pachycephala Rech.f.* (*Gharibi et al., 2019*), *Brassica napus* (*Rezayian, Niknam & Ebrahimzadeh, 2018*), *Arabidopsis* (*Nakabayashi et al., 2014*), grape (*Degu et al., 2015*; *Savoi et al., 2016*), and white clover (*Ballizany et al., 2012*). The physiological and molecular mechanisms underlying the drought-induced accumulation of these compounds to modulate phenylpropanoid as well as the flavonoid biosynthetic pathway need to be further elucidated by integrated transcriptome and metabolite profiling.

The monoterpenoid biosynthesis was significantly modulated by the prolonged and severe field drought conditions in the present experiment. Plant terpenes were synthesized in the plastids through the 2C-methyl-D-erythritol-4-phosphate pathway (MEP), and in the cytosol through the mevalonate (MVA) (*Tholl, 2006*). A number of terpenoid metabolites were involved in adaptation to adverse environments (*Pichersky & Raguso, 2018*; *Murphy & Zerbe, 2020*), including biotic and abiotic stresses; however, the knowledge of drought-modulated regulatory mechanism of monoterpene biosynthesis is limited (*Zhang et al., 2019*). All of the four-terpene synthase (TPS) genes encoding salutaridine reductase (SalR) and nerolidol synthase 1 involved in monoterpene biosynthetic pathway were downregulated under drought conditions. Salutaridine reductase catalyzes the stereo specific reduction of salutaridine to 7(S)-salutaridinol, nerolidol synthase 1 converts geranyl diphosphate (GPP) into S-linalool, and farnesyl diphosphate (FPP) into (3S)-E-nerolidol in the biosynthesis of morphin (*Ziegler et al., 2009*). Morphine resides within the diverse class of metabolites called benzylisoquinoline alkaloid, and drought stress, it has been noted, can increase alkaloids in opium poppy (*papaver soniniferum*) (*Szabó et al., 2003*). Our results were unlike previous findings in several plants (*Selmar & Kleinwächter, 2013*), those such as *Chrysopogon zizanioides* (*Ziegler et al., 2009*) and grapevine (*Griesser et al., 2015*; *Savoi et al., 2016*). *Ziegler et al. (2009)* reported upregulation of the gene encoding Salutaridine reductase under drought stress specifically in leaf tissue of *Chrysopogon zizanioides*. Drought-induced monoterpene production was observed in several plants (*Selmar & Kleinwächter, 2013*) including grapevine leaves (*Griesser et al., 2015*; *Savoi et al., 2016*). Six TPS genes, one of which included the nerolidol synthase 1-like gene, were differentially expressed in response to abiotic stresses in *Santalum album* (*Zhang et al., 2019*). Further biochemical and transcriptomic profiling is needed to address terpenoid biosynthetic pathways and their spatiotemporal regulation in response to adverse drought stress.

Transcription factors (TFs) modulate diverse transcriptional regulation and play significant regulatory roles in plant signaling responses to developmental and

environmental changes (*Ulker & Somssich, 2004*; *Osakabe et al., 2014*; *Yu et al., 2015*). In the present study, 4,438 differentially expressed TFs were identified to promote or suppress abiotic stress responses, including the bHLHs, NAC, MYB, ERF, C2H2s, C3Hs, and WRKY families. WRKY TFs have been reported to be involved in drought stress responses through the ABA signaling pathway (*Ulker & Somssich, 2004*; *Osakabe et al., 2014*). Overexpression of *ZmWRKY58* enhances the drought and salt tolerance in transgenic rice (*Cai et al., 2014*). Drought-responsive WRKY TFs *TaWRKY33* and *TaWRKY1* confer the transgenic *Arabidopsis* plants drought and/or heat resistance (*He et al., 2016*). The cotton WRKY TF *GhWRKY33* reduces transgenic *Arabidopsis* resistance to drought stress (*Wang et al., 2019*). In the present study, a total of 79 WRKY genes induced by field drought treatment were grouped in cluster 1, the majority of which were upregulated. *Li, Xu & Huang (2016)* reported 637 transcription factors responsive to dehydration in pear, among which 45 WRKY genes were differentially expressed. *Huang et al. (2015)* classified a total of 103 WRKY TFs in the pear genome, demonstrating an improvement of tolerance to drought by manipulating the PbWRKYs. Therefore, WRKY TFs may play significant roles in regulating drought stress responses.

## CONCLUSION

We utilized deep sequencing technology to investigate the transcriptome profiles in pear leaves, branches, and young fruits in response to the prolonged field drought induced by irrigation withdrawal. A total of 819 DEGs were detected, and 4,438 TFs were differentially expressed between drought and irrigation control, presenting valuable information on transcriptome changes in response to drought. We illustrated the flavonoids and monoterpenoid biosynthesis-related genes specifically expressed in drought and irrigation control during field-grown season in pear. Validation of gene expression by 13 randomly selected genes was in correspondence with transcriptomic results. Several candidate genes including flavonoid and terpenoid genes, transcription factors, and drought-responsive elements, were involved in transcriptional regulation of plant response to drought. Such information is important to germplasm management and in endeavoring to improve pear productivity.

### Funding

This work was supported by the Shanxi Province Natural Science Foundation (201801D121255, 20210302124122), the Research Subject of Agricultural Science and Technology Innovation of Shanxi Academy of Agricultural Sciences (YCX2018D2YS14), the Project of Scientific and Technological Innovation research of Shanxi Academy of Agricultural Sciences (YCX2020SJ10), and the China Agriculture Research System (CARS-28-28). There was no additional external funding received for this study. The funders had no role in study design, data collection and analysis, decision to publish, or preparation of the manuscript.

## Grant Disclosures

The following grant information was disclosed by the authors:

The Shanxi Province Natural Science Foundation: 201801D121255, 20210302124122.

The Research Subject of Agricultural Science and Technology Innovation of Shanxi Academy of Agricultural Sciences: YCX2018D2YS14.

The Project of Scientific and Technological Innovation research of Shanxi Academy of Agricultural Sciences: YCX2020SJ10.

The China Agriculture Research System: CARS-28-28.

## Competing Interests

The authors declare there are no competing interests.

## Author Contributions

- Sheng Yang conceived and designed the experiments, performed the experiments, analyzed the data, prepared figures and/or tables, authored or reviewed drafts of the paper, and approved the final draft.
- Mudan Bai performed the experiments, analyzed the data, prepared figures and/or tables, authored or reviewed drafts of the paper, and approved the final draft.
- Guowei Hao performed the experiments, analyzed the data, authored or reviewed drafts of the paper, and approved the final draft.
- Huangping Guo and Baochun Fu conceived and designed the experiments, authored or reviewed drafts of the paper, and approved the final draft.

## Data Availability

The sequence data is available at NCBI: PRJNA655255. The raw data are available in the Supplemental Files.

## Supplemental Information

Supplemental information for this article can be found online at http://dx.doi.org/10.7717/peerj.12921#supplemental-information.

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
