# Peer review of "Transcriptomics analysis of field-droughted pear (Pyrus spp.) reveals potential drought stress genes and metabolic pathways"

_PeerJ, doi:10.7717/peerj.12921_

## Round 0.1 · original submission · Major Revisions

Dear Dr. Baochun Fu and co-authors,

As you will see, our reviewers found that your work was interesting and important. However, they raised several comments to improve the manuscript.

Reviewer 1 suggests that the authors should identify some important metabolites associated with flavonoids and monoterpenoid biosynthesis. (S)he also had a concern about the format of Reference section.

Reviewer 2 provided minor essential comments. For example, adding references related to qRT-PCR method and software. This reviewer also had a concern about the number of validated gene expressions by qRT-PCR.

Reviewer 3 provided several essential comments. For example, there were concerns about the fold-change threshold, the RNAs used, and the content and legends of the Tables. In addition, most of the reviewers point out the need to polish the main text.

I would like to ask you to address or to respond with reasons not to follow the comments and suggestions made by these reviewers.

Best regards,
Atsushi Fukushima

Reviewer 1 ·

Basic reporting

It is ok

Experimental design

it is ok

Validity of the findings

it is ok

Additional comments

This work carries out a field-based transcriptomic study,and reveals potential drought stress genes and metabolic pathways.For the writing and results of the manuscript,I have some comments as follow.
1. This is necessary to improve logic in language expression.The manuscript needs extensive editing.
2. The flavonoids and monoterpenoid biosynthesis-related secondary metabolites were needed to identify.
3. The references are in a very confusing format. Authors should check this part very carefully.

Reviewer 2 ·

Basic reporting

The manuscript has many minor grammatical errors. These should be checked properly.

Experimental design

No comment

Validity of the findings

No comment

Additional comments

1. Line 96: Please re-write the sentence. Don’t write as “we provide a transcriptomic study….”. You may write it as- The major aim of the study was…..
2. Line 101: ‘Plant materials and growth conditions’ sub-headings should be changed. You have indicated location, plant materials, and experimental setup and data recordings of the experiment. So, please improve the sub-headings or concise the writings if you wish to keep the original ones.
3. Line 190 : 2 -ΔΔCT method please add reference.
4. Line 191-192: Pearson’s correlation software package with reference should be included.
5. Add “Conflict of interest” section in the text.
6. Arrange the references according to the Journal’s guideline.
7. Overall, Justify the text and follow the journal’s guideline.
8. Figure 3c, 3d and 4 not clear please increase resolution.
9. Results can be improved
10. Conclusions must be improved.
11. A total of A total of 1318 DEGs, and 4438 TFs were identified in present study, but only 13 genes were validated by qRT-PCR. It is necessary to use more genes for validation.

Reviewer 3 ·

Basic reporting

The manuscript by Yang et al reports the results of a global transcriptome analysis of the field drought response of pear, Pyrus spp. The document is well written. The Introduction presents the basic aspects of the response of plants to drought and the importance of the pear fruit crop. The research question and the structure of the article is well defined.

Experimental design

The methodology is well described and the results are clear. The authors use five replicates of both treatments, control and drought, mixing RNA from leaves, branches and young fruits. This is a strong point of this study. In general, Figures are relevant, well labelled and described. I have some comments about the content and legends of the Tables, which are indicated below.

Validity of the findings

The results of this transcriptome analysis identify novel pear genes differentially expressed between drought and irrigation field treatment. They can be considered interesting candidates to be followed up in a pear breeding program. I have not been able to find a link for the raw data. I guess it must be in some repository, but is not indicated in the manuscript.

Additional comments

I have some comments and suggestions about the main document, as follows:
L21-23: I suggest including a statement of the wide range of tissues sampled, i.e. leaves, branches and young fruits. This is an important point of this work.
L54: Instead of ‘deficient’ it should be ‘water deficiency’.
L134-135: RNA integrity was not measured with a Nanodrop, is it an Agilent Bioanalyzer?
L151-153: Equal amount of RNA from the different tissues was mixed prior to library construction. Can you differentiate the tissue-type later?
L161: The threshold used, |log2FoldChange>1| is quite liberal. It would be more appropriate to get a higher, stricter threshold, as highly expressed genes are those with (log2FoldChange < -3.5).
L178 and following: Was the same RNA used for RNA-seq and the qRT-PCR validation of gene expression? This should be specified in the document.
L197: CK is used as abbreviation for control treatment. It is correct but misleading as this term suggests cytokinin.
L204: Those numbers, 10 cDNA libraries (5 repeats for drought and irrigation) were sequenced, should appear in M&M.
L247-248: The statement: ”The second largest group (Figure 5b) contained 293 genes whose expression increased under field drought conditions”, does not agree with the Figure shown. Expression is only slightly higher under drought.
L275-276: Which genes of those shown in Figure 7 are downregulated? It is not apparent from the Figure or the legend.
L300: The sentence mentions 11 randomly selected DEGs, whereas 13 DEGs appear in M&M and Figure 4. Please check the numbers.
L358: Typo, delete ‘in’.
L384-385: The previous sentences talk about WRKY transcription factors. The current sentence does not follow from that. It should be conditional tense, WRKY transcription factors may play important roles in regulating drought stress responses.
L393: How many genes were validated by qRT-PCR, 11 or 13?
Figure 5: There are no differences in expression among the genes clustered in the Figure. They have a similar expression in all the treatments, flat.
Figures 7 and 8: What genes are differentially expressed of those shown?
Table 3: The legend mentions “Highly expressed genes identified in drought samples”, however it should specify “Highly expressed genes in control conditions, down-regulated under drought”. Is C control and D drought?
Table 4: The legend mentions “Highly expressed genes identified in control samples”. As before, I think it should be “Highly expressed genes identified under drought”. C control, D drought?
Table 5: The legend mentions “Top 10 GO terms of DEGs in field drought samples compared to CK”. Indeed, those are the top 10 GO terms for each of the three GO categories. It could be clarified.
Table 7: What is the log2FoldChange to declare significant a difference? Most ot the genes that appear in the table have low values, probably not significantly different.

---

## Round 0.2 · Major Revisions

Dear Dr. Baochun Fu and co-authors,

Thank you for your revision. I looked at your Rebuttal (filename: 10212_Response_Letter.docx) and was not able to find any point-by-point responses to all the comments raised by the reviewers in previous round of reviews.

Could you please check it and tell me the situation?

Best regards,
Atsushi Fukushima

---

## Round 0.3 · Minor Revisions

Dear authors,

Thank you for your revision. However, Reviewer 3 still had some comments to improve the manuscript. Would you please revise the manuscript?

Best regards

Reviewer 2 ·

Basic reporting

No Comment

Experimental design

No Comment

Validity of the findings

No Comment

Additional comments

Dear Authors, Thank you for improving your article, and I am agreed with the changes.

Reviewer 3 ·

Basic reporting

I still have some questions and comments to improve the manuscript.
L138: 5 repeats for drought and irrigation ‘each or respectively’.
L139: Diluted to ‘1 ng μL’ for library construction – It should be ng µl-1 or ng/µL
L224: “Genes that were highly expressed in CK samples …” Instead of CK it is better to use ‘irrigated or control’ samples, as has already been changed throughout the document.
L239-241 and 246-247: The same idea is expressed in both sentences, a group of genes with increased expression under drought. Looking at Figure 5, the profile of the average expression values in group 1 (A) and group 2 (B) is similar, why are they different groups?
L314-315: “A total of 1318 DEGs were detected”. Previously, in the Results section (L216), it was mentioned that “In total, 819 DEGs between drought and irrigation were identified”. The numbers do not match.
L392: “A total of 1318 DEGs were detected”. As mentioned in the previous comment, in the Abstract and Results, the number of DEGs is 819. Please check.
Figure 2: The control, irrigated treatment appears as CK.
Figure 4 legend: In the figure RNAseq and qRT-PCR results for the displayed genes are shown. Both of them should appear in the legend.

Experimental design

No comment

Validity of the findings

No comment

Additional comments

The current version of the manuscript integrates most of the comments or suggestions of the reviewers. The document has clearly improved and it reads much better now.

---

## Round 0.4 · Major Revisions

Dear authors,

Thank you for your revision. Our Section Editor had the following comments. Would you please revise the manuscript?

Best regards
Atsushi Fukushima

--
Our Section Editor has commented and said:

"There is absolutely no data attached to this manuscript, except for descriptions of what may exist. There is nothing presented to the reader to even begin to validate what is claimed. The style in which proposed genes are mainly hypothetical. Even if annotations have been provided there is no links to them.

Journal manuscripts are often scanned by text-mining software that locates and extracts core data elements, like gene function. Adding standard ontology terms, such as the Gene Ontology (GO, geneontology.org) or others from the OBO foundry (obofoundry.org) can enhance the recognition of your contribution and description. This will also make human curation of literature easier and more accurate. Tables would need to be constructed which would attach the annotations in both textual and descriptive forms to given sequences. None of this was visible.

The sequence data needs to be deposited in a third-party resource such as NCBI GenBank. Assembled transcriptomes can be deposited as a transcriptome shotgun assembly (GenBank TSA resource). Please see: https://www.ncbi.nlm.nih.gov/nuccore/
There appears to be about 40000 sequences which are unaccounted for and there is no link to any assemblies which would constitute the sequences of study.

The manuscript is currently in need of major revisions.

In the Discussion thread there were suggested changes to text; however, major revision is recommended to compensate for the lack of connectivity to the data."

Reviewer 3 ·

Basic reporting

No comment

Experimental design

No comment

Validity of the findings

No comment

Additional comments

The authors made the suggested corrections in the document and addressed all my queries. No more comments.

---

## Round 0.5 · accepted · Accept

Dear authors,

Thank you for your revision.

Best regards